# Application of Oxford Nanopore Technology to Plant Virus Detection

**DOI:** 10.3390/v13081424

**Published:** 2021-07-22

**Authors:** Lia W. Liefting, David W. Waite, Jeremy R. Thompson

**Affiliations:** Plant Health and Environment Laboratory, Ministry for Primary Industries, P.O. Box 2095, Auckland 1140, New Zealand; David.Waite@mpi.govt.nz

**Keywords:** biosecurity, ONT, MinION, diagnostics, plant viruses, high-throughput sequencing, rRNA depletion

## Abstract

The adoption of Oxford Nanopore Technologies (ONT) sequencing as a tool in plant virology has been relatively slow despite its promise in more recent years to yield large quantities of long nucleotide sequences in real time without the need for prior amplification. The portability of the MinION and Flongle platforms combined with lowering costs and continued improvements in read accuracy make ONT an attractive method for both low- and high-scale virus diagnostics. Here, we provide a detailed step-by-step protocol using the ONT Flongle platform that we have developed for the routine application on a range of symptomatic post-entry quarantine and domestic surveillance plant samples. The aim of this methods paper is to highlight ONT’s feasibility as a valuable component to the diagnostician’s toolkit and to hopefully stimulate other laboratories towards the eventual goal of integrating high-throughput sequencing technologies as validated plant virus diagnostic methods in their own right.

## 1. Introduction

In 1989, while thinking about synthetic lipid membranes, David Deamer sketched his idea of an artificial pore through which DNA would thread its way, altering the ionic current in a manner that was specific for each nucleotide base. Twenty-three years later, Deamer’s dream was showcased by Oxford Nanopore as being able to sequence forty-eight kilobases of genomic DNA on both strands in one pass through a protein pore [1]. This ongoing technological revolution in nucleic acid high-throughput sequencing (HTS) was given an early roadmap with the “The $1000 Genome” project which challenged the research community with the goal of developing high-throughput single long-read methods that would cost less than $1000 USD to sequence a mammalian-sized genome with 99.99% accuracy [2]. At present, Oxford Nanopore Technologies (ONT) are as close to any in achieving this goal, with reads up to 2.3 billion bases in length [3] and a single routine run costing $1000 USD. Other long-read technologies such as Pacific Biosciences are equally as attractive but like ONT have lower accuracy when compared to synthesis-based technologies (e.g., Illumina, cPAS). At its inception, the raw read accuracy of ONT was approximately 85% but recently, with improvements in base-calling algorithms, a consensus accuracy of greater than 99.9% has been reported [4,5]. These ongoing reductions in sequence error rates combined with its relatively cheap capital costs and portability make the ONT MinION sequencer a particularly attractive platform for lower-throughput applications. The Flongle, a MinION adapter, extends these specifications yet further, with a less powerful (2.8 Gb vs. 50 Gb) but even more cost-effective platform ($90 vs. $1000).

Despite its potential, the application of ONT for plant virus detection appears to be still in its infancy, with only two relevant peer-reviewed publications in 2020; this is out of a total of 89 publications on ONT in virology in the same year (Figure 1).

In the past four years, there have been only nine ONT plant virus publications recorded in the National Center for Biotechnology Information Pubmed database on approaches ranging from genotyping viruses in potato [6] and yam [7] to plant virus surveillance [8] and whole-genome sequencing [9].

Viruses can significantly and unexpectedly disrupt our way of life as they constantly evolve and adapt to new environments. Assuming human activities return to pre-COVID-19 days, their distribution around the world, principally aided by us, will continue to challenge the growing global infrastructure. In the case of plant viruses, they represent close to half of all emerging plant pathogens and can significantly impact agricultural production [10]. An increasing global human population combined with climate change will provide added stress to the food supply chain requiring ever more sophisticated methods to ensure regional biosecurity [11].

Established methods for virus diagnostics of germplasm can still involve long and laborious methods such as biological indexing [12]. HTS technologies offer the diagnostician the tools to screen propagative plant material for any potential pathogen without a priori knowledge of the pathogen’s genetic composition. This, in theory, provides a faster alternative to biological indexing, which it is hoped, in the long run, will expedite germplasm movement through quarantine hubs [13,14,15].

Within this objective, our activities at the Ministry for Primary Industries Plant Health and Environment Laboratories (PHEL) are involved in developing the application of new technologies such as ONT for enhanced pathogen detection, with the eventual aim of method validation and harmonization with other regulatory institutions around the world. Recently, in this same Special Issue on the “Pros and Cons in the Use of Next-Generation Sequencing for Plant Virus Diagnosis”, Mehetre et al. [16] provided an excellent overview on the developments and challenges in plant viral diagnostics including an update on ONT, where they concluded that it is the “most readily applicable to (plant) viral diagnostics”.

Here, we provide step-by-step details of a protocol that we now routinely employ using the ONT MinION sequencing device with Flongle flow cells for diagnostic testing of symptomatic post-entry quarantine and domestic surveillance samples. This method has been successfully used on a whole range of different plant species to detect both viruses and viroids as well as phytoplasmas and liberibacters and is a now considered an essential method in our diagnostic toolkit.

## 2. Method Specifics

The overall method can be split into two main activities: the wet and the dry lab. The wet-lab work can be completed in a full day, while the dry-lab or bioinformatics part can take three to four days depending on the amount of data obtained (Figure 2).

The current ONT library preparation kits for RNA samples are designed for polyadenylated (poly(A)) RNA and unfortunately viruses that lack a poly(A) tail, a description which includes many plant viruses as well as viroids, will not be sequenced with these preparation strategies. These preparation kits include the direct cDNA sequencing kit (SQK-DCS109), the PCR-cDNA sequencing kit, with (SQK-PCB109) and without (SQK-PCS109) barcoding, and the direct RNA sequencing (SQK-RNA002). To sequence all types of RNA for generic virus and viroid detection, two strategies were compared using citrus tristeza virus (CTV)-infected *Citrus*. CTV is a phloem-limited virus of the genus *Closterovirus* with a 19.3 kb single-stranded positive-sense RNA genome that lacks a poly(A) tail. The first strategy involved performing a poly(A)-tailing reaction on the total RNA using *Escherichia coli* Poly(A) Polymerase (New England Biolabs, Cat. No. M0276) according to the manufacturer’s instructions and then using the resulting poly(A) RNA as input in the ONT direct cDNA sequencing kit (SQK-DCS109). In the second strategy, double-stranded (ds) cDNA was synthesised using random hexamers and used as input in the end-prep step in the direct cDNA sequencing kit (SQK-DCS109).

The libraries were run separately on MinION flow cells. Following quality filtering, the library prepared from the poly(A)-tailing reaction produced 461,686 reads, whereas the library prepared from random primed ds cDNA produced 1,041,002 reads. For comparison of the two datasets, a subset of 460,000 reads from each dataset were mapped to the reference genome of CTV (GenBank Accession No. NC_001661), resulting in 125 reads mapping from the poly(A)-tailing reaction and 540 reads mapping from the random primed ds cDNA (Table 1).

The lower number of reads that mapped to CTV from the poly(A)-tailing reaction may be due to incomplete poly(A)-tailing of the RNA fragments, especially the viral sequences which make up a small proportion of the total fragments. Most of the reads from the poly(A)-tailing reaction mapped to the 3′ end of the virus (Figure 3A), whereas the reads from the library using random primed ds cDNA mapped evenly across the reference sequence, resulting in complete coverage of the virus genome (Figure 3B). Wongsurawat et al. (2019) [17] also observed the same phenomenon, although the 3′ bias was not as extreme.

Based on these results, it was decided to use random primed ds cDNA for library preparation. However, random primers used for cDNA synthesis have the potential to anneal at many random points on an RNA transcript creating short fragments of cDNA and therefore full-length transcripts are not sequenced with our choice of methodology. This negates the long-read advantage of ONT, but we deem it more advantageous to obtain complete coverage of the virus genome and believe that this trade-off is acceptable. The length of the cDNA reads depends on the amount of random primer used. Generally, the more primer added, the shorter the cDNA, but interestingly, the dataset generated by random primed ds cDNA shown in Table 1 has a very similar mean read length to that generated by poly(A)-tailed RNA. Subsequent datasets have a shorter mean read length due to a lower input of total RNA. The concentration of random primer was not reduced to avoid a reduction in cDNA yield.

Flongle flow cells yield sufficient data for our purposes. However, protocols for the Flongle flow cell are currently not available from ONT for the RNA and cDNA library preparation kits. For this work, we modified the MinION protocol from the end-prep step by performing half size reagent volumes. For increased efficiency, libraries for two Flongle flow cells are prepared and sequenced simultaneously.

### Host Ribosomal RNA Depletion

Ribosomal RNAs (rRNAs) are extremely abundant, constituting 80–90% of total RNA. The removal of plant host rRNAs (ribodepletion) prior to HTS is essential to enrich for pathogens present in a sample. Although numerous kits are available for ribodepletion of human, mouse and rat, limited kits are available for plants. Efficient ribodepletion of plant RNA requires the incorporation of extra probes for mitochondrial and chloroplast rRNA removal. A recently developed plant ribodepletion kit for plants is the QIAseq FastSelect from Qiagen (https://www.qiagen.com/us/products/discovery-and-translational-research/next-generation-sequencing/rna-sequencing/ribosomal-rna-and-globin-mrna-removal/qiaseq-fastselect-rrna-plant-kits/?clear=true#orderinginformation, accessed on 21 July 2021) which prevents the synthesis of cDNA from plant rRNA. The FastSelect reagent is combined with total RNA and incorporated into the reverse-transcription step during the library preparation workflows for Illumina sequencing kits. For application of the FastSelect ribodepletion kit into our ONT workflow, we initially followed the QIAseq FastSelect protocol for the NEBNext Ultra II Directional Library Prep Kit where the FastSelect reagent is incorporated with the random primers during the ds cDNA synthesis step. This protocol resulted in a low yield of the final library (average of 0.44 ng/µL from 6 samples). The inclusion of a purification step with AMPure XP beads between the QIAseq FastSelect and ds cDNA synthesis steps increased the final library yield (average of 1.6 ng/µL from 45 samples).

## 3. MinION Method for Plant Viruses (Wet Lab)

### 3.1. Materials

#### 3.1.1. Equipment

Benchtop centrifuge;Quick spin centrifuge;Nanodrop spectrophotometer (or equivalent);Qubit fluorometer (or equivalent);Thermocycler;Heat block or water bath;Vortex mixer;Hula mixer sample mixer (e.g., ThermoFisher, Cat. No. 15920D);P1000, P200, P20, P10, P2 pipettes;Magnetic separation rack for 1.5 mL tubes;ONT MinION sequencing device;ONT Flongle adapter.

#### 3.1.2. Reagents

Nucleic acid extraction kit (e.g., RNeasy Plant Mini Kit (Qiagen Cat. No. 74,903 or 74904), InviMag Plant DNA Mini Kit (Invitek Cat. No. 7437300250));RapidOut DNA Removal Kit (Thermo Scientific, Cat. No. K2981);QIAseq FastSelect—rRNA Plant Kit (Qiagen, Cat. No. 334311, 334315, 334,317 or 334319);5X First-Strand Buffer (Thermo Scientific Y02321) supplied with SuperScript III Reverse Transcriptase (Thermo Scientific 18080044, 18,080,093 or 18080085) (or similar);Agencourt AMPure XP beads (Beckman Coulter Cat. No. A63880, A63881, or A63882);Maxima H Minus Double-Stranded cDNA Synthesis Kit (Thermo Scientific, Cat. No. K2561, K2562 or K2563);GeneJET PCR Purification Kit (Thermo Scientific, Cat. No. K0701 or K0702);NEBNext Ultra II End Repair/dA-Tailing Module (New England Biolabs, Cat. No. E7546);NEB Blunt/TA Ligase Master Mix (New England Biolabs, Cat. No. M0367);ONT Direct cDNA Sequencing Kit (Cat. No. SQK-DCS109);ONT Flow Cell Priming Kit (Cat. No. EXP-FLP002)—provided with sequencing kitONT Flongle Sequencing Expansion (Cat. No. EXP-FSE001)—provided with Flongle flow cells;Qubit RNA HS Assay Kit (Thermo Scientific Cat. No. Q32852 or Q32855);Qubit 1X dsDNA HS Assay Kit (Thermo Scientific Cat. No. Q33230 or Q33231)Ethanol;Isopropanol;Nuclease-free water (e.g., Thermo Scientific Cat. No. R0582).

#### 3.1.3. Consumables

ONT Flongle flow cells;1.5 mL tubes;1.5 mL LowBind tubes;0.2 µL PCR tubes;P1000, P200, P20, P10, P2 pipette filter tips.

### 3.2. Procedure

#### 3.2.1. Nucleic Acid Extraction (1 h—If No Ethanol Precipitation Requirement)

(a)Grind approximately 0.5 g of symptomatic tissue in 5 mL of homogenization buffer.(b)Extract total RNA or total nucleic acid using standard protocols, for example, Qiagen RNeasy Plant Mini Kit or the InviMag Plant DNA Mini Kit on a Kingfisher automated workstation.(c)Extract several replicates for each sample in case of a low yield.(d)Determine the approximate nucleic acid concentration using a Nanodrop spectrophotometer.(e)If the sample concentration is less than ~50 ng/µL, concentrate by ethanol precipitation.

NOTE: Although the host ribodepletion step requires that the input RNA is at a concentration of 32.3 ng/µL, a higher concentration than this is needed to allow for inaccuracies from Nanodrop quantification and slight dilution of the sample during the DNA removal step.

STOPPING POINT: The nucleic acid can be stored short term at −20 °C or long term at −80 °C.

#### 3.2.2. DNA Removal (40 min)

(a)Add the components listed in Table 2 into a 1.5 mL tube (the reaction can be scaled down to 10 µL maintaining constant ratios of components).

(b)Vortex gently or mix by pipetting and spin down.(c)Incubate at 37 °C for 30 min.(d)Remove the DNase I by adding 2 µL of DNase Removal Reagent (DRR) for each microlitre of DNase I used. Before use, vortex the DRR until completely resuspended.(e)Incubate at room temperature for 2 min gently mixing 2–3 times to resuspend the DRR.(f)Centrifuge the tube at >800× *g* for 1 min to pellet the DRR.(g)Immediately transfer the supernatant into a new tube. Take care not to transfer any of the DRR.(h)Quantitate the RNA using a Qubit fluorometer.

STOPPING POINT: The RNA can be stored short term at −20 °C or long term at −80 °C.

#### 3.2.3. Host Ribodepletion (1 h)

(a)Add the components listed in Table 3 into a 0.2 mL PCR tube on ice.

(b)Mix gently by pipetting and spin down.(c)Place the tube(s) in a thermocycler once it has reached 75 °C and incubate as described in Table 4.

(d)Transfer the sample to a 1.5 mL LoBind tube. Add 32 µL (0.8X) of resuspended AMPure XP beads. Mix well by pipetting up and down at least 10 times and spin down.(e)Incubate at room temperature for 5 min.(f)Place the tube in a magnetic separation rack. After the solution is clear, slowly remove and discard the supernatant, being careful not to disturb the beads.(g)Add 200 µL of freshly prepared 80% ethanol to the tube while in the magnetic rack. Wait for 30 s, then carefully remove and discard the ethanol.(h)Repeat the previous step for a total of 2 washing steps. Completely remove all traces of ethanol after the second wash.(i)Air dry the beads for 5 min while the tube is in the magnetic rack. Do not over dry.(j)Elute the RNA from the beads with 15 µL of nuclease-free water. Pipette mix 10 times and spin down.(k)Incubate at room temperature for 2 min.(l)Place the tube in a magnetic separation rack until the solution is clear.(m)Remove 14 µL of the supernatant into a new PCR tube and place on ice.(n)Quantitate the ribodepleted RNA using a Qubit fluorometer.

EXPECTED YIELD: 400–800 ng total.

STOPPING POINT: The nucleic acid can be stored short term at −20 °C or long term at −80 °C.

#### 3.2.4. Double-Stranded cDNA Synthesis (2.5 h)

First-strand cDNA synthesis

(a)Add the components listed in Table 5 into a 0.2 mL PCR tube on ice.

(b)Mix gently by pipetting, spin down and incubate at 65 °C for 5 min. Chill on ice, spin down again and place on ice.(c)Add the components listed in Table 6 in the indicated order.

(d)Mix gently by pipetting, spin down and incubate at 25 °C for 10 min, followed by 50 °C for 30 min.(e)Terminate the reaction by heating at 85 °C for 5 min, then place on ice.(f)Continue immediately with the second-strand synthesis reaction.

Second-strand cDNA synthesis.

(g)Add the components listed in Table 7 in the indicated order to give a total volume of 100 µL.

(h)Mix gently by pipetting and spin down.(i)Incubate at 16 °C for 60 min.(j)Stop the reaction by adding 6 µL of 0.5 M EDTA, pH 8.0 and mix gently.(k)Remove residual RNA by adding 10 µL of RNase I to the second-strand synthesis reaction tube and incubate at room temperature for 5 min.

Double-stranded cDNA purification using the Thermo Scientific GeneJET PCR Purification Kit

(l)Transfer the cDNA (116 µL) to a 1.5 mL tube and add an equal volume of Binding Buffer (116 µL). Mix thoroughly.(m)Add a 1:2 volume of 100% isopropanol (116 µL). Mix thoroughly.(n)Transfer the solution from step m to the GeneJET purification column. Centrifuge at top speed for 1 min. Discard the flow-through and place the purification column back into the collection tube.(o)Add 700 µL of wash buffer to the purification column. Centrifuge at top speed for 1 min. Discard the flow-through and place the purification column back into the collection tube.(p)Centrifuge the empty purification column for an additional 1 min to completely remove any residual wash buffer.(q)Transfer the purification column to a new 1.5 mL LoBind tube. Add 30 µL of Elution Buffer (prewarmed to 65 °C) to the centre of the column membrane and incubate for 1 min. Centrifuge at top speed for 1 min.(r)Pipette the eluate to the centre of the same column membrane and incubate for 1 min. Centrifuge at top speed for 1 min. Discard the purification column.(s)Quantitate the cDNA using a Qubit fluorometer.

EXPECTED YIELD: The ONT MinION protocol for the direct cDNA sequencing kit (SQK-DCS109) states to start the end-prep step with 70–200 ng of cDNA. As we are using half size reagent volumes for Flongle flow cells, 35–100 ng cDNA is required. On average, we obtain approximately 40 ng in total from ds cDNA synthesis. Frequently, amounts less than 35 ng are used, and good sequence outputs have been obtained from inputs as low as 15 ng.

STOPPING POINT: The double-stranded cDNA can be stored indefinitely at −20 °C or −80 °C.

#### 3.2.5. End-Prep (50 min)

(a)Perform end repair and dA-tailing of cDNA by mixing the components listed in Table 8 into a 0.2 mL PCR tube.

(b)Mix gently by pipetting and spin down.(c)Incubate at 20 °C for 5 min and 65 °C for 5 min.(d)Transfer the sample to a new 1.5 mL LoBind tube, add 30 µL of resuspended AMPure XP beads to the end-prep reaction and mix by pipetting.(e)Incubate on a Hula mixer at room temperature for 5 min.(f)Spin down the sample and place the tube in a magnetic separation rack. After the solution is clear, slowly remove and discard the supernatant, being careful not to disturb the beads.(g)Add 200 µL of freshly prepared 70% ethanol to the tube while in the magnetic rack. Carefully remove and discard the ethanol.(h)Repeat the previous step for a total of two washing steps.(i)Spin down and place the tube back on the magnet. Pipette off any residual ethanol. Allow to dry for ~30 sec, but do not dry the pellet to the point of cracking.(j)Remove the tube from the magnetic rack and resuspend the pellet in 23 µL of nuclease-free water. Incubate at room temperature for 2 min.(k)Place the tube in a magnetic separation rack until the solution is clear.(l)Remove 22.5 µL of the supernatant into a new 1.5 mL LoBind tube.

STOPPING POINT: It is recommended to take the end-prepped cDNA immediately into adapter ligation, but if necessary, it can be stored for several days at −20 °C.

#### 3.2.6. Adapter Ligation (1 h)

(a)Perform adapter ligation of the end-prepped cDNA by assembling the reaction mix described in Table 9, mixing by flicking the tube between each sequential addition.

(b)Mix gently by flicking the tube and spin down.(c)Incubate at room temperature for 10 min.(d)Add 20 µL of resuspended AMPure XP beads to the adapter ligation mix and mix by pipetting.(e)Incubate on a Hula mixer at room temperature for 5 min.(f)Place the tube in a magnetic separation rack until the solution is clear and pipette off the supernatant.(g)Add 100 µL of wash buffer (WSB) to the beads and resuspend by pipetting to remove free adapter. Return the tube to the magnetic separation rack, allow the beads to pellet and pipette off the supernatant.(h)Repeat the previous step.(i)Spin down and place the tube in the magnetic separation rack. Pipette off residual supernatant.(j)Remove the tube from the magnetic separation rack and resuspend in 7 µL of Elution Buffer (EB).(k)Incubate on a Hula mixer at room temperature for 10 min.(l)Pellet the beads on a magnet until the eluate is clear.(m)Remove and retain 7 µL of eluate into a clean 1.5 mL LowBind tube.(n)Quantitate 1 µL of eluted cDNA using a Qubit fluorometer.(o)Store the library on ice until ready to load the Flongle flow cell.

EXPECTED YIELD: On average, we obtain approximately 1.6 ng/µL and occasionally obtain yields as high as 5 ng/µL and load the entire amount onto the Flongle flow cell. Good sequence outputs have been obtained from library concentrations as low as 0.29 ng/µL. One of our largest datasets obtained to date of 1,180,000 reads had a library concentration of 0.57 ng/µL.

#### 3.2.7. Priming and Loading the Flongle Flow Cell (20 min)

Note: In order to improve Flongle flow cell performance, the Flush Buffer, Sequencing Buffer II, and Loading Beads II are provided in glass vials in the Flongle Sequencing Expansion kit that is shipped with the Flongle flow cell order.

(a)Perform a Flongle flow cell check with the MinKNOW software. Keep the flow cell in the MinION device for priming and loading of the flow cell.(b)In a new 1.5 mL LoBind tube, mix 117 µL of Flush Buffer (FB) with 3 µL of Flush Tether (FLT) and mix by pipetting. Use this mix to prime the Flongle flow cell according to the protocol on the Oxford Nanopore website.(c)Prepare the sequencing mix in a new 1.5 mL LowBind tube as described in Table 10.

(d)Immediately load the sequencing mix into the Flongle flow cell according to the protocol on the ONT website.

## 4. Dataset Analyses (Dry Lab)

### 4.1. Materials

Note: Version numbers reflect the versions used during our analysis and are not recommendations to use the specific version over any other, particularly where newer software releases are available.

#### 4.1.1. Software

MinKNOW (version 21.06.0, https://nanoporetech.com, accessed on 21 July 2021).MinKNOW will install a CPU-only version of the guppy base-calling tool when installed.We recommend that the GPU-enabled version of guppy be used where possible. However, this version is only supported for Linux-based operating systems and must be obtained from the ONT website and manually installed after MinKNOW.

Software (for command line analysis).

BLAST+ (version 2.10.0, https://ftp.ncbi.nlm.nih.gov/blast/executables/blast+/LATEST/, accessed on 21 July 2021).seqmagick (version 0.7.0, https://fhcrc.github.io/seqmagick/, accessed on 21 July 2021) or seqtk (version 1.3, https://github.com/lh3/seqtk, accessed on 21 July 2021).minimap2 (version 2.17, https://github.com/lh3/minimap2, accessed on 21 July 2021).samtools (version 1.10, https://github.com/samtools/samtools, accessed on 21 July 2021).EFetch (part of the Entrez Programming Utilities, version 13.3, https://www.ncbi.nlm.nih.gov/books/NBK25501/, accessed on 21 July 2021).

Software (for GUI-based analysis, optional).

Geneious Prime (version 2021.1.1, https://www.geneious.com/, accessed on 21 July 2021).

#### 4.1.2. Equipment

Sequencing is performed using a Dell Precision 3640 tower with several hardware modifications to facilitate real-time base-calling. These are as follows:1 TB solid state drive (SSD) for read/write operations with MinION Flongle outputs. The SSD has been configured with a logical partition to separate the operating system from the MinION output data to safeguard against disk space errors.4 TB hard disk drive for storing data following sequencing, to maximise the free space on the SSD.GeForce RTX 2070 SUPER graphics processing unit to facilitate GPU-accelerated base-calling. There are many options when selecting a suitable GPU for performing base-calling, but currently only Nvidia (https://www.nvidia.com/, accessed on 21 July 2021) GPUs are supported by ONT. The model selected for our workflow can perform high accuracy (HAC) base-calling of a full MinION sequencing output with guppy version 5.0.7 in approximately 30 min and is easily capable of performing base-calling for multiple, simultaneous Flongle runs.A power supply unit (PSU) capable of handling the increased power draw from the GPU. All graphics processors from the Nvidia RTX family require additional power supply beyond what is supplied directly through motherboard and this may be more than the default PSU can provide. When selecting a GPU, the minimum and recommended power requirements can be found through the Nvidia website and we strongly encourage interested parties to research this with care.

GPU accelerated base-calling is only supported on Linux operating systems so our sequencing computer uses the Ubuntu 18.04.5 LTS operating system. Following sequencing and real-time base-calling, classification of reads is performed using the New Zealand eScience Infrastructure (NeSI) high performance compute cluster to run a BLASTn search against a local copy of the NCBI nucleotide (nt) database. This database is updated approximately every 6 months to ensure that the reference is current.

### 4.2. Procedure

#### 4.2.1. Performing Sequencing and Live Base-Calling (24 h)

(a)Connect the MinION device to the computer and set up the sequencing run as appropriate for the library type.Enable real-time base-calling, using the high-accuracy (HAC) model.Enable quality filtering, rejecting low quality reads. For guppy version 4.2.2 the default quality threshold is Q < 7, but this has increased to Q < 9 for guppy 5.0.7/5.0.11.

(b)Set an appropriate run time for the device.24 h is sufficient to obtain sufficient reads, depending on the quality of the flow cell and library.After this time there will be very few active pores remaining on the flow cell.

(c)When finished select the run of interest and save the run report, using the ‘Export PDF’ option from the experiment view.

#### 4.2.2. Convert FASTQ Files to FASTA Format (Geneious Prime) (5 min)

(a)Open Geneious Prime and use the batch import tool (File -> Import -> Files…) to import all contents of the ‘fastq_pass’ folder.Enable to ‘Create sequence list’ option.Note: Geneious Prime may attempt to format your data as paired-end sequencing. Be sure to provide the correct sequencing platform and read orientation if prompted.

(b)Select all imported files and use the context menu (right click) to select ‘Group sequences into a list…’(c)Select the newly created sequence list and export the file in FASTA format (File -> Export -> Documents…). Set the “Files of Type” option to “FASTA sequences/alignment (*.fasta)”.

#### 4.2.3. Convert FASTQ Files to FASTA Format (Linux Command Line) (5 min)

(a)Use the cat command to copy the contents of each individual FASTQ file into a single output file:cat fastq_pass/*.fastq > all_sequences.fastq.

(b)Use a tool such as seqmagick or seqtk to convert the FASTQ output to FASTA format. The FASTQ sequences produced through guppy contain metadata in the sequence description and although some analysis tools, such as DIAMOND, will ignore these data BLAST+ does not. It is advisable to create a copy of your data with metadata removed for simplicity of the downstream analysis, e.g.,seqmagick convert --first-name all_sequences.fastq all_sequences.fasta.seqtk seq -A -C all_sequences.fastq > all_sequences.fasta.

#### 4.2.4. Initial BLASTn Search (12 h)

(a)Perform a BLASTn search of the sequences against the NCBI nt database.When specifying the output format, select one of the tab-delimited tables formats (i.e., ‘-outfmt 6′ or ‘-outfmt 7′) to make follow-up analysis much easier.Add additional fields to make it easier to sight-inspect the annotation results. Good options are ‘salltitles’ (titles), ‘sscinames’ (scientific names), ‘scomnames’ (common names), or ‘staxids’ (NCBI Subject Taxonomy ID).Note: Running on the NeSI platform with 16 CPUs typically requires 9 h to complete classification for one sequencing library. However, this number varies according to the number of sequences obtained and our longest BLAST job required 37 h to complete.For the first pass, keep the results reasonably strict, limit the maximum number of target sequences per query, and only accept matches with a high percentage identity to reference sequences. As an example, for our first round of classification, we will require sequence identity of at least 90%, an e-value of less than 1e-3, and only take the top three matches per query sequence.

(b)Add column names to your file to aid analysis if the BLASTN output format does not do this for you (for example, outfmt 6).(c)Before commencing with analysis, we use a screening script to remove any sequence hits to plant or vertebrate sequence. This is an optional step to make the resulting datasets more amenable to searches and manipulations.(d)Tables can be inspected in any tool for working with spreadsheets (e.g., Microsoft Excel, R, Tableau).(e)Filter the list of hits down to organisms of interest (e.g., virus, viroid, phytoplasma or liberibacter species). This can be performed manually, or through retaining a list of known species names or NCBI taxonomy identifiers (taxid) to use as search parameters.

#### 4.2.5. Curation of BLASTn Results and Follow-Up Analysis (1–4 h)

(a)For each hit, or groups of hits, to an organism of interest inspect the NCBI website to ensure that the target sequence is correctly annotated, particularly if the result is to an environmental (clone) sequence rather than an isolate.(b)If the target sequence appears to be correctly annotated, obtain a copy of the reference sequence and perform sequence mapping.efetch -format fasta -db sequences -id [NCBI_ACCESSION] > [NCBI_ACCESSION].fna.minimap2 -ax splice NCBI_ACCESSION.fna all_sequences.fastq | samtools view -F 4 > NCBI_ACCESSION.sam.

(c)Visualise the mapping alignment to confirm that these results could not be explained by mapping to a conserved or low-complexity region of the target. This is best performed in Geneious Prime but if necessary, a text file containing the per-position mapping depth can be produced using samtools then visualised using any software with plotting capabilities.samtools sort NCBI_ACCESSION.sam > NCBI_ACCESSION.bam.samools depth -a NCBI_ACCESSION.bam > NCBI_ACCESSION.map_depth.txt.

(d)Additional confirmatory sequence queries can be made to further examine the initial search results. Two such options are:For a particular mapping file of interest, extract the mapped reads and assemble using a tool such as Geneious Prime, Canu, or Flye. The assembly can then be subjected to a new BLASTn query to ensure that the result is correct.For viruses of interest which encode protein sequences (i.e., not viroids), translated protein searches using DIAMOND in blastx mode can be used to extend the classification process.

## 5. Case Studies

The ONT method described in this manuscript is now routinely used by PHEL for generic screening of symptomatic plants for viruses and virus-like organisms. To date, over 50 samples from a wide range of plant genera have been run on Flongle flow cells, including *Acer*, *Actinidia*, *Camellia*, *Fragaria*, *Lathyrus*, *Pinus*, *Prunus*, *Rosa*, and *Vitis*, producing an average output of 680,000 reads per sequencing run. Viruses from a range of genera as well as viroids and a liberibacter have been detected. Details of the sequencing run for three of these samples are provided below.

### 5.1. New Host Association

A kiwifruit (*Actinidia* sp.) sample was received showing chlorotic and necrotic spots on the leaves (Figure 4A). ONT sequencing on a Flongle flow cell produced a total of 320,130 reads and an estimated 100 Mb with an average read length of 279 bp. After base-calling the 296,046 passed reads were subjected to BLASTn analysis against the nt database and 278 reads had significant identity to tomato spotted wilt virus (TSWV). Mapping of the reads to the three genomic segments of TSWV, RNA M, RNA S, and RNA L (GeneBank Accession Numbers NC_002050, NC_002051, and NC_002052, respectively) resulted in 89 reads mapping with a mean coverage of 2.5 (Figure 4B). The presence of TSWV in the sample was confirmed by a specific qPCR assay. This is the first report of TSWV infecting *Actinidia* worldwide which highlights the power of this technology in detecting unexpected host pathogen associations.

### 5.2. Mixed Virus Infection

A sweet pea (*Lathyrus odoratus*) sample showing symptoms of chlorotic mottle and streaks on the leaves was run on a Flongle flow cell producing a total of 829,320 reads and an estimated 719 Mb with an average read length of 807 bp. After base-calling the 804,564 passed reads were subjected to BLASTn analysis against the nt database. Approximately 49,000, 11,000, and 564,000 reads produced significant identities to alfalfa mosaic virus (AMV), bean yellow mosaic virus (BYMV), and white clover mosaic virus (WClMV), respectively. The reads were mapped to the reference sequences of these three viruses producing high genome coverage (Table 11).

These three viruses are widespread in sweet pea worldwide. However, this is the first record of BYMV in sweet pea in New Zealand. Many viruses infect sweet pea and using ONT sequencing on this sample highlights the efficiency achieved in being able to detect all viruses in a single sequencing run rather than performing many individual PCR assays.

### 5.3. Unusual Virus-Like Symptoms

A strawberry (*Fragaria* × *ananassa*) sample was received from a strawberry runner grower showing symptoms of pink to purple colouration of the leaves. ONT sequencing on a Flongle flow cell produced a total of 304,350 reads and an estimated 136 Mb with an average read length of 411 bp. When the 248,974 passed reads were subjected to BLASTn against the nt database, no viruses or viroids were identified in the dataset. Surprisingly 31 reads had significant identity to *Liberibacter crescens* and 6 reads to ‘*Candidatus* Liberibacter africanus’ or ‘*Candidatus* Liberibacter asiaticus’. When the reads were mapped to the complete genome sequence of *Liberibacter crescens* (GenBank Accession Number NZ_CP010522), 47 reads mapped with all but 2 of them mapping to the 16S or 23S rRNA genes. Identification of liberibacters are based on sequence analysis of their 16S rRNA gene. Mapping the reads to the *Liberibacter crescens* 16S rRNA gene (GenBank Accession Number NR_102476) produced a 337 bp consensus sequence from the strawberry liberibacter that was not of sufficient length or quality to identify the liberibacter to species level. Conventional PCR on the symptomatic strawberry sample using a combination of universal and liberibacter-specific 16S rRNA primers and Sanger sequencing of the amplicons produced a near-complete 16S rRNA sequence. The liberibacter detected in strawberry appears to be a previously undescribed species of liberibacter and further work is in progress to confirm this finding. Liberibacters are phloem-limited bacteria with circular genomes of approximately 1.15 to 1.52 Mb. Apart from *Liberibacter crescens*, liberibacters are unable to be cultured in vitro and are classified according to the candidatus species concept. Liberibacters have been associated with several diseases including citrus and solanaceous plants. However, this is the first record of a liberibacter infecting strawberry. Using ONT sequencing on this sample highlights the ability of this technology to detect pathogens present in a low titer.

## 6. Conclusions

HTS using the ONT MinION sequencing device has greatly improved the accuracy and efficiency of plant virus diagnosis at PHEL, with the added benefit that it will also detect viroids, phytoplasmas and liberibacters in the same run. We have shown that ONT sequencing is especially useful in detecting mixed infections, new host associations and previously undescribed species. Sample preparation using the random primed ds DNA method has enabled generic detection of different pathogen types. The cost to run a single sample on a Flongle flow cell, including ribodepletion and ds cDNA synthesis is approximately $220 USD. The wet-lab activities can be performed with minimal training by personnel with molecular biology laboratory skills. Instructional videos are available on the ONT website for flow cell loading which inexperienced users find to be the most difficult part of the procedure. Up to four samples can be easily processed simultaneously until the end of the ds cDNA purification step, and two samples at a time for the Flongle library preparation. Data analyses are most efficiently performed through the Linux command-line and experience working within such an environment is recommended. Access to high-performance computers for BLAST searches greatly reduces the time required to perform analyses. The ability of ONT to sequence the complete genome of viruses depends largely on the host–virus combination. In the examples we have shown, there was a great variation in the sequence coverage of the viruses detected in the host. For TSWV in kiwifruit, a large part of the genome was not sequenced and a mean coverage of only 2.5 resulted in a low-quality consensus sequence. In contrast, sweet pea infected with three different viruses resulted in high depth coverage of their complete genomes. We are using ONT as a generic screening tool and if necessary, the presence of viruses and other pathogens are confirmed by PCR. If a greater sequence depth is required, for example, for new species or strains, the sample is sequenced on an Illumina platform. Although ONT sequencing using Flongle flow cells has proven effective at detecting pathogens in symptomatic samples, we have not yet determined whether more than one sample can be multiplexed on the same run or whether it is sensitive enough to test that asymptomatic plants in quarantine are free of viruses and viroids.

## Figures and Tables

**Figure 1 viruses-13-01424-f001:**
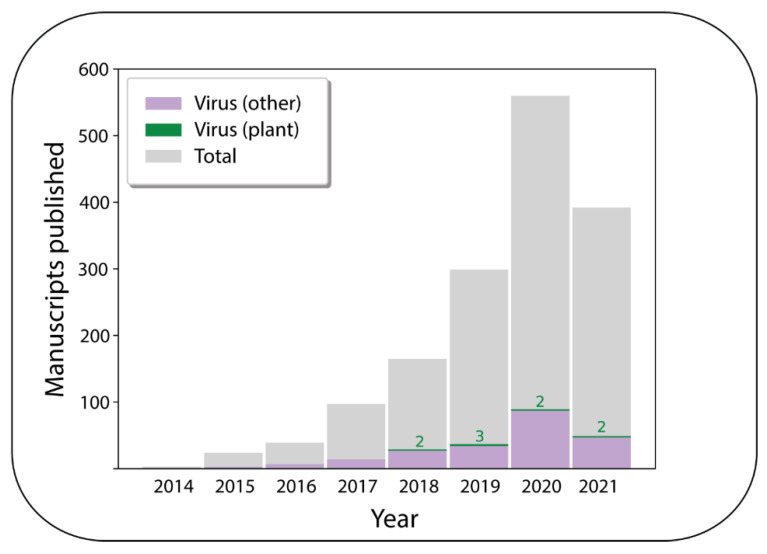
A histogram showing the number of peer-reviewed publications each year relating to Oxford Nanopore Technologies’ MinION platform as registered in the National Center for Biotechnology Information Pubmed database. The numbers in green represent the number of these publications that studied plant viruses. The database search was performed on the 10 June 2021 using the keywords “nanopore”, “nanopore sequencing”, and “nanopore minion”. After manual curation of the results for unrelated manuscripts and review/perspective articles, a total of 3896 publications were identified.

**Figure 2 viruses-13-01424-f002:**
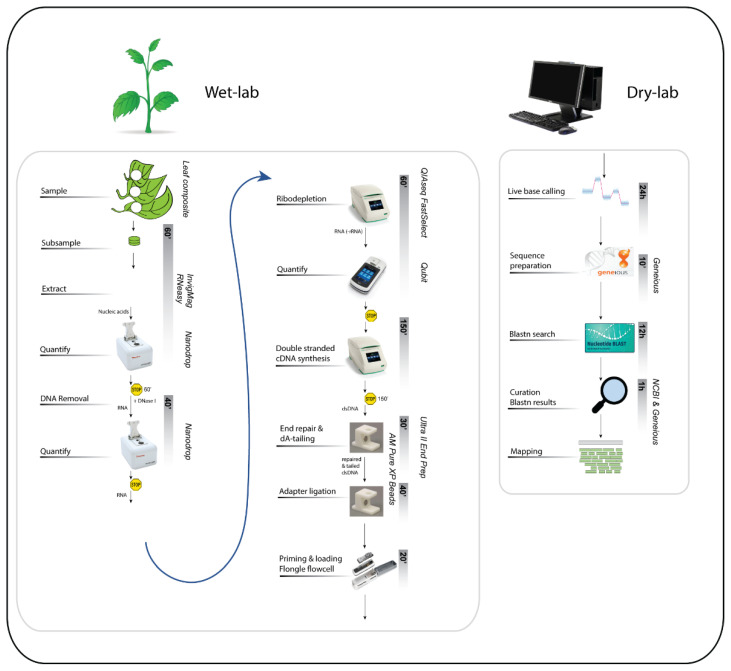
An overview of the main steps of the ONT Flongle plant virus method from wet lab to dry lab. Each activity is listed on the left, with options to pause indicated by the yellow “stop” sign. The estimated duration of each activity is shown within the grey vertical bars. Detailed descriptions of each step are provided below in the Procedure.

**Figure 3 viruses-13-01424-f003:**
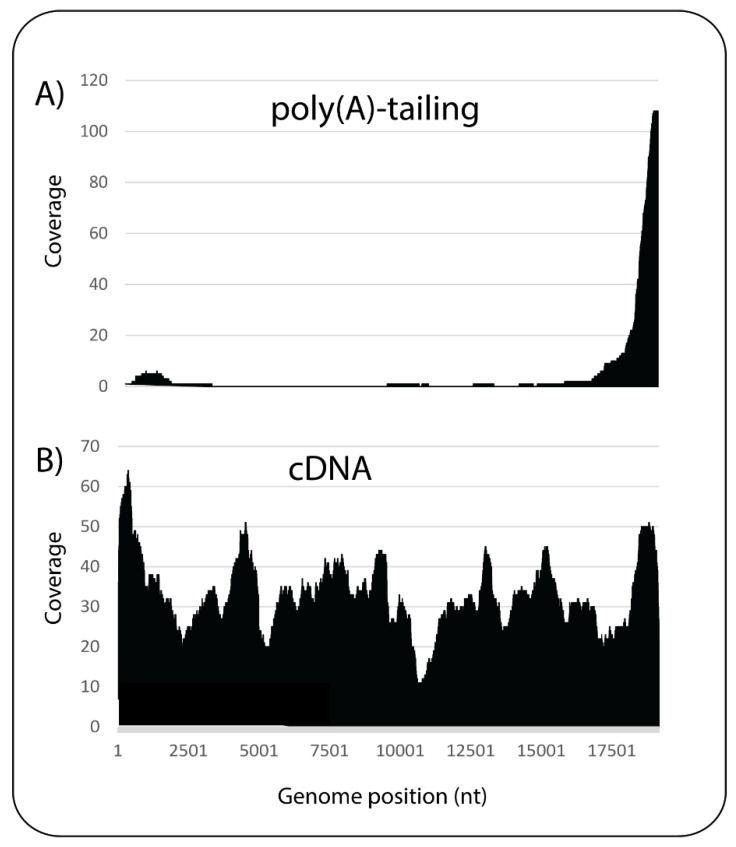
Read coverage plots mapping to citrus tristeza virus, comparing the poly(A)-tailing method (**A**) and the random primed ds cDNA method (**B**). Numbers on the x-axes refer to genome nucleotide (nt) position.

**Figure 4 viruses-13-01424-f004:**
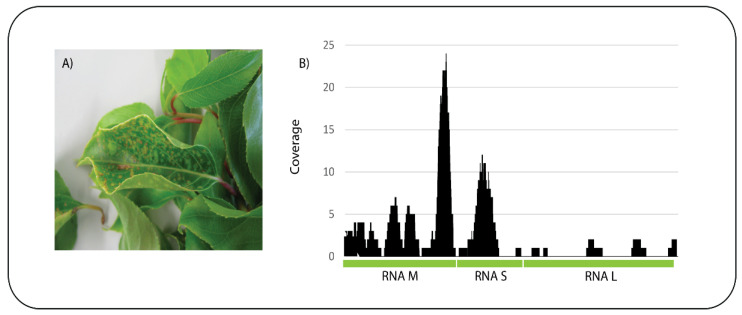
*Actinidia* showing symptoms of chlorotic and necrotic spots on the leaves (**A**) and sequence reads from the sample mapped to the three concatenated genome segments (RNA M, S and L) of tomato spotted wilt virus (**B**).

**Table 1 viruses-13-01424-t001:** Read and mapping summary of a subset of 460,000 reads generated from libraries prepared by either poly(A)-tailing reaction or from random primed double-stranded (ds) cDNA of citrus tristeza virus (CTV)-infected total RNA.

Sequencing Strategy	Read Length (bp)			Reads Mapped to CTV	Depth of CTV		
	Mean	Minimum	Maximum		Mean	Minimum	Maximum
Poly(A)-tailing reaction	1106	86	48,827	125	8.2	0	108
Random primed ds cDNA	1187	120	55,079	540	33.4	7	64

**Table 2 viruses-13-01424-t002:** DNA removal reaction mix.

Component	Volume
RNA sample (<20 µg)	85 µL
10× DNase buffer with MgCl2	10 µL
DNase I	5 µL

**Table 3 viruses-13-01424-t003:** Host ribodepletion reaction mix.

Component	Amount or Volume
Total RNA (DNA removed)	1 µg
5X First-Strand Buffer *	8 µL
QIAseq FastSelect—rRNA Plant	1 µL
Nuclease-Free Water	to 40 µL

* The 5X First-Strand Buffer comes with the Invitrogen SuperScript III RT enzyme for PCR. Alternative buffers with a similar composition may be used.

**Table 4 viruses-13-01424-t004:** FastSelect hybridization protocol.

Temperature	Time
75 °C	2 min
70 °C	2 min
65 °C	2 min
60 °C	2 min
55 °C	2 min
37 °C	2 min
25 °C	2 min

**Table 5 viruses-13-01424-t005:** Random hexamer binding reaction mix.

Component	Volume
Ribodepleted RNA	13 µL
Random Hexamer	1 µL

**Table 6 viruses-13-01424-t006:** First-strand cDNA synthesis reaction mix.

Component	Volume
4X First-Strand Reaction Mix	5 µL
First-Strand Enzyme Mix	1 µL

**Table 7 viruses-13-01424-t007:** Second-strand cDNA synthesis reaction mix.

Component	Volume
First-Strand cDNA Synthesis Reaction Mixture	20 µL
Nuclease-Free Water	55 µL
5X Second-Strand Reaction Mix	20 µL
Second-Strand Enzyme Mix	5 µL

**Table 8 viruses-13-01424-t008:** End repair and dA-tailing reaction mix.

Component	Volume
Purified double-stranded cDNA sample	25 µL
Ultra II End-Prep Reaction Buffer	3.5 µL
Ultra II End-Prep Enzyme Mix	1.5 µL

**Table 9 viruses-13-01424-t009:** Adapter ligation reaction mix.

Component	Volume
End-prepped cDNA	22.5 µL
Adapter Mix (AMX)	2.5 µL
Blunt/TA Ligation Master Mix	25 µL

**Table 10 viruses-13-01424-t010:** Sequencing mix.

Component	Volume
Sequencing Buffer II (SQBII)	14 µL
Loading Beads II (LBII), mixed immediately before use	10 µL
cDNA library	6 µL

**Table 11 viruses-13-01424-t011:** Viruses identified in sweet pea by Oxford Nanopore sequencing showing the level of coverage.

Virus	GenBank Accession No.	Per-Nucleotide Mapping Depth		
		Mean	Minimum	Maximum
AMV	NC_001495 (RNA1)	3998	6	7935
	NC_002024 (RNA2)	1692	6	2888
	NC_002025 (RNA3)	3810	14	7919
BYMV	NC_003492	620	0	1329
WClMV	NC_003820	6958	52	8089

## Data Availability

All data generated during this study are included in this article.

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
