# Peer review of "Application of Oxford Nanopore Technology to Plant Virus Detection"

_viruses, 2021, doi:10.3390/v13081424_

Round 1

Reviewer 1 Report

The work "Application of Oxford Nanopore technology to plant virus detection" (ID viruses-1302238) is very interesting and proposes a new approach in the analysis of viruses in plants. I recommend publication with minor revision of the manuscript

Although it is specified in the indications of the extraction kit, it is good to specify in the manuscript the amount of plant material that is used for each nucleic acid extraction process at the beginning of the protocol.

In the paragraph “conclusion”:

-It would be interesting to know what the author think about the use of composite samples.

-What the authors think about the use of this technique in asymptomatic plant material (e.g. directly on imported plant material)?

-Indicate approximately, how many samples can be processed per person, using the proposed protocol.

Author Response

Reviewer 1

Although it is specified in the indications of the extraction kit, it is good to specify in the manuscript the amount of plant material that is used for each nucleic acid extraction process at the beginning of the protocol.

Authors response: This information has been included in the nucleic acid extraction section 3.1 (a), line 222.

In the paragraph “conclusion”:

-It would be interesting to know what the author think about the use of composite samples.

-What the authors think about the use of this technique in asymptomatic plant material (e.g. directly on imported plant material)?

-Indicate approximately, how many samples can be processed per person, using the proposed protocol.

Authors response: These points have been included in the conclusion section (lines 597-599 and 610-614).

Reviewer 2 Report

The manuscript form Liefting and co-authors present the use of Oxford Nanopore Technologies for the detection of viruses in plant samples.

As a plant and fungal virologist I found the protocol quite interesting and the possibility to use ONT for virus detection as well as for virus discovery is really important. 

I would ask to the authors some minor points:

  • Did you confirmed the viral infection using other techniques (PCR, northern blot, etc...)? In that case can you please highlight it?
  • I would suggest to add, at least in the introduction or in the conclusion section, some sentences on the possibility to directly sequence RNA. In this line I would ask to the authors if they tried to directly sequence RNA and if so what results they obtained. This is particularly interesting for several reason: as they mentioned in the introduction, often viruses don't have poly and/or the 5' cap. The use of ONT directly applied on RNA would overtake these limitations and would present the possibility to gain information about the RNA modification.

Author Response

Did you confirmed the viral infection using other techniques (PCR, northern blot, etc...)? In that case can you please highlight it?

Authors response: The identification of viruses and other pathogens from Oxford Nanopore data are confirmed by PCR in cases of new host associations, new species or low sequence coverage. This is stated in case studies 5.1 (lines 538-539) and 5.3 (lines 575-577) and in the conclusion (lines 607-609).

I would suggest to add, at least in the introduction or in the conclusion section, some sentences on the possibility to directly sequence RNA. In this line I would ask to the authors if they tried to directly sequence RNA and if so what results they obtained. This is particularly interesting for several reason: as they mentioned in the introduction, often viruses don't have poly and/or the 5' cap. The use of ONT directly applied on RNA would overtake these limitations and would present the possibility to gain information about the RNA modification.

Authors response: The direct RNA sequencing protocol from Oxford Nanopore is designed for polyadenylated RNA. This fact is stated in lines 96-101 of the manuscript.